

# Anticancer potential of ethanolic sidr leaf extracts against MCF-7 breast cancer cells: phytochemical, nutritional, and antimicrobial profile comparisons of different plant parts

Roqayah H. Kadi[1], Nashi K. Alqahtani[2,3], Ahmed M. Abdulfattah[4,5], Fayez Alsulaimani[4,5], Ahmed M. Basri[4,5], Reham M. Algheshairy[6], Hend F. Alharbi[6], Rokayya Sami[7], Amal Alyamani[8], Roua S. Baty[8], Ruqaiah I. Bedaiwi[9], Hala M. Abo-Dief[10], Nahid A. Osman[10], Nouf H. Alsubhi[11], Ashwaq M. Al-Nazawi[12,13] and Manal Malibary[14,15]

[1] Department of Biological Sciences, College of Science, University of Jeddah, Jeddah, Saudi Arabia
[2] Date Palm Research Center of Excellence, King Faisal University, Al Hofuf, Saudi Arabia
[3] Department of Food and Nutrition Sciences, College of Agricultural and Food Sciences, King Faisal University, Al-Ahsa, Saudi Arabia
[4] Embryonic Stem Cell Unit, King Fahd Medical Research Center, King Abdul Aziz University, Jeddah, Saudi Arabia
[5] Department of Medical Laboratory Sciences, Faculty of Applied Medical Sciences, King Abdulaziz University, Jeddah, Saudi Arabia
[6] Department of Food Science and Human Nutrition, College of Agricultural and Food, Qassim University, Buraydah, Saudi Arabia
[7] Department of Food Science and Nutrition, College of Sciences, Taif University, Taif, Saudi Arabia
[8] Department of Biotechnology, Taif University, College of Sciences, Taif, Saudi Arabia
[9] Department of Medical Laboratory Technology, Faculty of Applied Medical Sciences, University of Tabuk, Tabuk, Saudi Arabia
[10] Department of Science and Technology, University College-Ranyah, Taif University, Taif, Saudi Arabia
[11] Biological Sciences Department, College of Science & Arts, King Abdul Aziz University, Jeddah, Saudi Arabia
[12] Department of Public Health, College of Nursing and Health Sciences, Jazan University, Jazan, Saudi Arabia
[13] Laboratory Department, Jazan University Hospital, Jazan University, Jazan, Saudi Arabia
[14] Department of Food and Nutrition, Faculty of Human Sciences and Design, King Abdul Aziz University, Jeddah, Saudi Arabia
[15] Food, Nutrition and Lifestyle Unit, King Fahd Medical Research Center, King Abdul Aziz University, Jeddah, Saudi Arabia

Corresponding author
Rokayya Sami, rokayya.d@tu.edu.sa

## ABSTRACT

**Background.** Traditional medicine has long utilized natural plants to treat diseases and promote overall health. They have contributed significantly to the creation of modern medications. To supplement existing information, therefore, this current investigation aimed to understand the potential therapeutic properties of the sidr plant against triple negative breast cancer (TNBC) on MCF-7 breast cell line by conducting phytochemical, nutritional, and antimicrobial profile comparisons of different plant parts.
**Methods.** Phytochemical analyses involved soluble sugar composition and oxidation profile, whereas nutritional analysis involved the proximate chemical analysis. Additionally, antimicrobial analyses involved six kings of food-borne bacterial strains.

**Results**. According to the study, leaves possessed the highest amounts of protein (12.44%) and ash (8.17%), as well as the highest amounts of ascorbic acid and total chlorophyll. On the other hand, pulps exhibited the highest flavonoid concentration in their ethanolic extract and had higher sugar contents. Furthermore, at a dosage of 100 µg/mL, the ethanolic extract of leaves showed potent antimicrobial action and suppressed over 50% of MCF-7 breast cancer cell survival.

**Conclusions**. Bioactive components, antioxidant, antimicrobial and nutritional elements of sidr plant appears a promising medicinal candidate to fight breast cancer cells.

# INTRODUCTION

Medicinal plants and their metabolites or even phyto-compound derivatives have been used as traditional medicines to inhibit, release, and treat diseases since ancient periods (*Zahara et al., 2022*). Correlated to this laboratory work, several plant extracts were established into significant cancer chemotherapeutics (*Majrashi et al., 2023*). Cancer disease is considered the 2nd cause of disability and death accompanied by a giant socio-economic burden worldwide (*Alzehr et al., 2022*). Cancer is a multifactorial disease, which can be due to some genetic and epigenetic aspects leading to apoptosis with rapid cell senescence (*Kamian et al., 2023*). Over the past few decades, cancer treatments have improved with the advent of immunotherapy and other targeted cancer medications (*Al-Eisa et al., 2023*). In many cases, radiotherapy, chemotherapy, and surgery were the available treatments to improve survival rates (*Ahmed et al., 2022a*; *Ahmed et al., 2022b*). However, after remission, tumors may recur and become resistant to the standard therapies (*Ishfaq et al., 2022*). They can result in toxicity, anemia, alopecia, nausea, and harm healthy tissues (*Boogaard, Komninos & Vermeij, 2022*). As a result, research into innovative approaches, efficient cancer prevention, and treatments are needed to fortify the global public health systems (*Rokayya et al., 2013*). *Ziziphus spina-christi* (L.) is commonly well-known as sidr that primarily grown in arid and semi-arid areas which recognized as a member of Rhamnaceae family multipurpose plant (*Mesmar et al., 2022*). The pharmacological potentials, such as health advantages, nutrients, antioxidants, anti-inflammatory agents, phytochemical components, hepatoprotective, antinociceptive, antihypertensive, antidiabetic, and antibacterial activities, have long been recognized by several cultures (*Al-Naimi et al., 2023*). Natural sidr extracts are non-toxic and effective in decreasing the chemotherapy doses and controlling some of the metabolic targets (*Abdallah, 2017*). It was found that sidr extracts reduced the carcinogenic phenotype of HeLa human cervical cancer, sarcoma-180 (S-180), MCF-7 human breast cancer, OVCAR-3 ovarian, HT-29 colon, A-498 kidney cancer, K-562 leukemia, and Capan-2 human pancreatic cancer cells (*Abdallah et al., 2024*). Approximately 30% of women suffer breast cancer, surpassing lung cancer and cancer-related deaths according to the

*World Health Organization (2021)*. Triple-negative breast cancer (TNBC) is an aggressive disease, that has a great possibility to improve resistance to the therapy with scarcity of therapy alternatives (*So et al., 2022*).

The scientific assessment of the sidr plant's phytochemical composition and biological activities is still incomplete and insufficient, despite the plant's long history of traditional use in folk medicine across many countries. Without a thorough comparison across many anatomical components, the majority of earlier research has concentrated on a single plant portion, usually the leaves. The distribution of bioactive compounds in plant tissues, including leaves, pulp, and seeds, varies significantly, and this can have a big impact on pharmacological potential and therapeutic use.

Both conventional and contemporary methods of plant extraction are essential for separating bioactive substances for use in food, medicine, and agriculture (*Abdallah, 2017*). Because they are easy to use, inexpensive, and effective at extracting a variety of phytochemicals, traditional techniques like maceration, percolation, decoction, and Soxhlet extraction have been around for a while. Since techniques require no sophisticated equipment and enable efficient extraction using common solvents as ethanol, these techniques are still useful today, particularly in institutional and research contexts (*Salih et al., 2019*). With respect to its excellent solvating ability for polar and semi-polar molecules, non-toxicity, and adherence to green chemistry principles, ethanolic extraction is especially preferred among them. Traditional ethanol-based methods are still widely used, despite the speed and efficiency benefits of contemporary techniques such as pressurized liquid extraction (PLE), supercritical fluid extraction (SFE), microwave-assisted extraction (MAE), and ultrasound-assisted extraction (UAE) (*Abdallah et al., 2024*). In light of its affordability, regulatory acceptability, and compatibility with locally accessible plant materials, ethanolic extraction is becoming a more common practice in Saudi Arabian institutional chemistry labs.

The current investigation aimed to highlight the potential therapeutic properties of the sidr plant against TNBC on MCF-7 breast cell line with a comparative study for proximate chemical analysis, mineral profiling, physicochemical analysis, soluble sugar composition, oxidation profile, and antimicrobial assessments.

## MATERIALS AND METHODS

### Schematic overview of the experimental program

In order to demonstrate the sidr plant's potential therapeutic benefits against TNBC on the MCF-7 breast cell line, a comparative analysis of proximate chemical analysis, mineral profiling, physicochemical analysis, soluble sugar composition, oxidation profile, and antimicrobial assessments were conducted. Figure 1 summarizes the performed chemical tests and serious work conducted in phases. Four kg of the fresh sidr plant were collected at heavy rainfall in August–October from a local orchard named after the natural plant reserve in Taif City, KSA. The fruit was selected to avoid broken, shrunken fruits, any signs of microbial growth, or physical damage. While leaves were devoid of extraneous
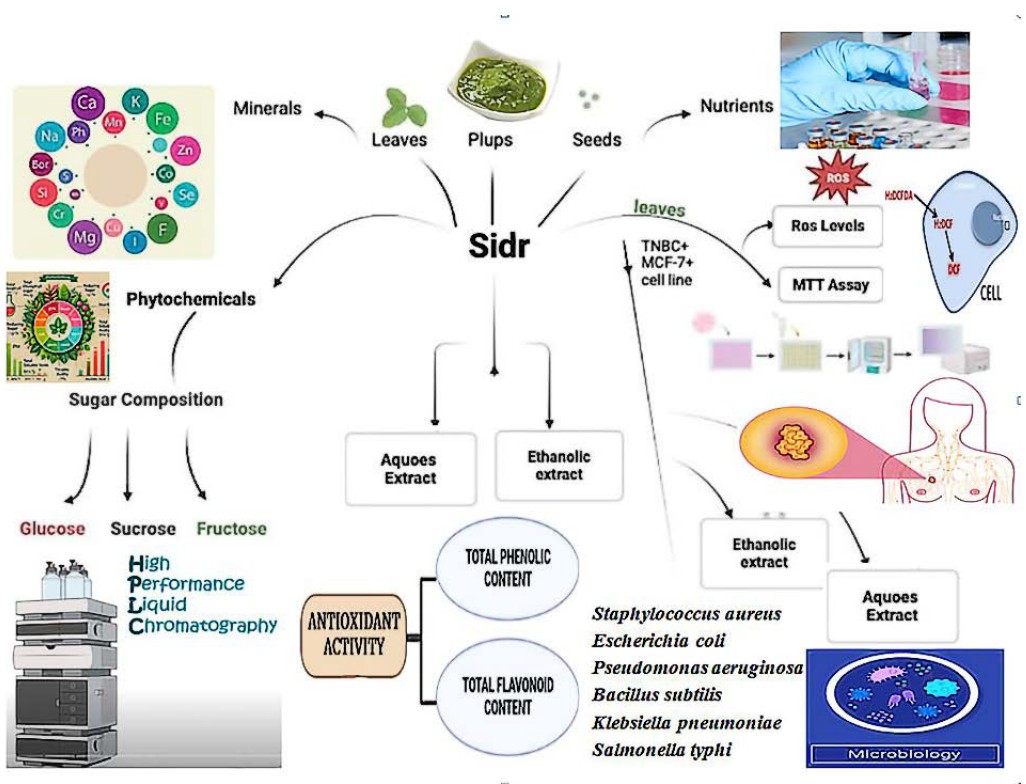

**Figure 1   The summary of the experimental work for sidr plant.**

objects and dust according to the identification by the Department of Biotechnology at Taif University.

## Preparation of plant samples

A sterile stainless-steel knife was used to manually separate the fresh fruits into pulps and seeds after they had been carefully washed with tap water. Fruit leaves, pulps, and seeds of *Ziziphus spina-christi* (L.) were dried at 45 °C for 15 to 20 h until reaching a constant weight in a thermostatic oven (UN750; Memmert, Schwabach, Germany). The dried samples were crushed into particles by using a milling machine (Deutsche CF-Hilfe e.V., Idstein, Germany) which might pass through 20 mesh screens, then sealed in an air-tight sterilized glass jar, and kept at 4 °C until further use (*Hashem & El-Lahot, 2021*).

## Preparation of aqueous and ethanolic extracts

The powdered samples were extracted using two different solvents according to the described protocol by *Guezzoun et al. (2024)*. Approximately 100 g of leaves, pulps, and seeds powder at a 10% w/v concentration were successfully submerged with a magnetic stirring in distilled water (one L) and ethanol (80%, one L) for 24 h at the ambient temperature. After the solvents underwent filtration, the extracts were vacuum-concentrated by using a rotary evaporator (R-200; Büchi, Zurich, Switzerland) at a temperature ≤45 °C. After being moved to a sample vial, the aqueous and ethanolic

extracts were stored in the dark at 4 °C until additional examination. Figure 1 summarizes the performed chemical tests and serious work conducted in phases.

## Proximate chemical analysis

Moisture, ash, ether, protein, fat, and fiber for leaves, pulps, and seeds were detected according to the Association of Official Analytical Chemists (*Association of Official Analytical Chemists (AOAC), 1995*). A hot air oven was used to dry the fresh samples (Baker's Pride, TN, USA) at 110 °C until a consistent weight was achieved to determine the moisture contents. Following drying, the sample's weight difference was noted on percentage of the dry weight (DW). Dried samples were burned for five hours at 500 °C–550 °C in a muffle furnace to detect ash percentage. By weighing the thimble containing the samples before and after extraction, the percentage of the ether extract was determined by the weight loss. Total nitrogen content was calculated using the Kjeldahl technique to evaluate the protein percentage by multiplying ($N \times 6.25$) by the elemental analyzer (FP628; LECO). Using the Fibrebag technique (Gerhardt, UK), the samples were digested with 0.13 M sulfuric acid and 0.313 M sodium hydroxide to determine the fiber percentage. The Soxhlet extractor (Gerhardt, Brackley, UK) was used to analyze the percentage of fats by using petroleum ether at 42 °C to 62 °C for 6 h. Carbohydrate contents were detected by difference by detracting the sum of moisture, ash, fat, and protein from 100% (*Association of Official Analytical Chemists (AOAC), 2000*).

## Mineral profiling

Mineral contents were estimated for ash residues according to (*Association of Official Analytical Chemists (AOAC), 2019*). Eleven elements were detected including calcium (Ca), phosphorus (P), sodium (Na), potassium (K), magnesium (Mg), zinc (Zn), copper (Cu), manganese (Mn), iron (Fe), chromium (Cr), and boron (B). The absorbance was detected by a spectrometer (ICP-OES, 2380, England). Results were expressed in (mg/100 g).

## Physicochemical analysis

The total chlorophyll for various sidr parts was detected according to the acetone extraction (80%) after incubation for 15 min in the dark then centrifuging at 2,500 rpm for 3 min and absorption at 645 nm (*Wang et al., 2022*). Results were expressed as (mg/g). Reducing sugar and starch was also detected according to the protocol (*Association of Official Analytical Chemists (AOAC), 2007*). The acidity (pH) was estimated by a pH meter (Mettler Toledo EL20). The total soluble solids (TSS) were detected by a refractometer (Model PAL-3; Atago) and expressed in Brix°. Titratable acidity (TA) was estimated by the titration assay against the malic acid, while results were expressed as % (*Association of Official Analytical Chemists (AOAC), 1995*). Ascorbic acid was detected by 2,6-dichlorophenolin-dophenol dye as described before in *Association of Official Analytical Chemists (AOAC) (2007)*, while results were expressed in (mg/100 g).

## Soluble sugar composition

High-performance liquid chromatography (HPLC) (LC-10AD, Shimadzu, Japan) was used to detect the sugar's composition. Standards of sucrose, fructose, and glucose were

purchased from Sigma-Aldrich (St. Louis, MO, USA). Approximately 0.5 g of various sidr parts were extracted in five mL of ethanol-water (90:10, V: V), and heated in a water bath for 30 min at 80 °C, then centrifuged at 6,000 rpm for 10 min. The supernatants were collected and adjusted to 25 mL and filtered by using a membrane size (0.22 μm). An LC-NH2 column (250 mm × 4.6 mm, five μm) with a flow rate (one mL/min) and 10 μL of injection volume were used to separate sugar compositions at 40 °C (*Gao et al., 2012*).

## Oxidation profile

Several assays for phytochemicals and antioxidant activities were detected on 500 μL of aqueous and ethanolic extracts. Total phenolics were detected by spectrophotometry by Folin-Ciocalteu reagent (10%) at 25 °C, while results were expressed as gallic acid equivalent (mg GAE/g). Pure water was used as a blank, while samples were evaluated at 765 nm (*Begum et al., 2024*). The total flavonoid was evaluated by using the aluminum chloride complexation processes (10%) and was expressed as quercetin equivalent (mg QE/100 g). The yellowish-orange color's intensity was detected spectrophotometrically at 415 nm using distilled water as a blank (*Benabderrahmane et al., 2024*). The antioxidant activities were detected by using methanol as a blank with two different methods as the stable free radical 2,2-diphenyl-1-picrylhydrazyl (DPPH) as a reagent and $IC_{50}$ as the amount of antioxidant concentration needed to decrease 50% of DPPH radicals by using a linear concentration/percentage inhibition curve (*Taha et al., 2024*). Using spectrophotometry, the intensity of the yellow hue was determined at 517 nm. As a reference standard, ascorbic acid was also used, while results were expressed as percentages and (mg/ml), respectively (*Uguru et al., 2023*).

## Antimicrobial assessments

The agar well-diffusion assay was applied to evaluate the antimicrobial assessments of the aqueous and ethanolic extracts (*Ahmed et al., 2022a*; *Ahmed et al., 2022b*). Six kings of food-borne bacterial strains—against *Staphylococcus aureus, Escherichia coli, Pseudomonas aeruginosa, Bacillus subtilis, Klebsiella pneumonia*, and *Salmonella typhi* were procured from Agriculture's cultural in Jeddah, KSA. The strains were stocked and cultured overnight at 37 °C in a nutritional broth medium (Merck, Darmstadt, Germany). A sterile pasture pipette was used to drill six mm wells in Muller-Hinton Agar medium (MHA; Merck). The medium was taken in powder form, liquefied, and then formed into a two mm layer of agar in a sterile petri dish, which was then stored at the ambient temperature resulting in a lawn culture of the bacteria (*Khojah, Abdelhalim OB & Habib, 2022*). A sensitivity test was conducted using microorganisms as standard organisms that had been diluted in a (0.9%) normal saline solution. The examined bacteria were then added by using a micropipette after a sample of equal volumes of bacterial cell suspension (0.1 ml) for each extract, at a concentration of 0.2 mg/ml, was placed into each well. Approximately 10 mg/L of tetracycline was used as a positive control, while distilled water and 80% ethanol served as the negative controls (*Sarjito et al., 2021*). Following an incubation period of 15 min, the extracts were incubated upright at 37 °C for 24 h after waiting an hour to diffuse into the agar substrate. The antibacterial assessments of the sidr extracts for all bacteria strains were

assessed by determining the zone of inhibition's diameter in (mm) by using a translucent ruler.

## Anticancer activity of sidr leaf extracts
### Cell culture
The Shanghai Institute of Biological Sciences in China was the source of the human breast cancer cell line (MCF-7). The cells were cultured in RPMI-1640 media and 10% DMEM supplemented with 150 units/mL of penicillin-streptomycin and 10% fetal bovine serum (Sigma-Aldrich). The cells were grown in a humidified chamber at 37 °C with 5% $CO_2$ condition (*Qiao et al., 2020*).

### Cell viability
Using a mitochondrial-dependent reduction of 3-(4,5-dimethylthiazol-2-yl)-2,5-diphenyl tetrazolium bromide (MTT) test (Sigma-Aldrich), mitochondrial respiration—a measure of cell viability—was ascertained. The MCF-7 cells were then subjected to various concentrations (0, 10, 25, 50, and 100 µg/mL) against the cells treated with the vehicle (equal ethanol concentration), whose viability was taken to be 100% for 24 h. In 96-well plates, treated MCF-7 cells with the ethanolic leaf extracts ($1 \times 10^5$ cells/ml) were incubated with MTT (five mg/ml) for 4 h before being dissolved in dimethyl sulfoxide (150 µl/well) with the blue dye and shaken for 15 min. The absorbance was detected at 550 nm after washing twice to evaluate the degree of MTT decrease inside the cells after changing the yellow day to purple color in the living MCF-7 cells by a microplate reader (HT, BioTek, USA) (*Rokayya et al., 2013*). Cell viability as a percentage was evaluated based on the concentrations of the ethanolic leaf extracts.

## Reactive oxygen species
Cells were labeled with a 2,7-dichlorfluorescein-diacetate (DCFH-DA) kit to evaluate changes in intracellular reactive oxygen species (ROS) production. The MCF-7 cells were incubated for the entire night in 6-well culture plates in a $CO_2$ incubator. MCF-7 cells were incubated for 30 min at 37 °C with 10 mol/L DCFH-DA after being treated for 48 h with the ethanolic leaf extracts at various concentrations as (0, 10, 25, 50, and 100 µg/mL). Following treatment with one µL, the cells in the positive control group were labeled with DCFH-DA and cultured for 1 h at 37 °C. Following collection, washing, and resuspension in phosphate buffer saline, cells were analyzed for fluorescence intensity at 485–535 nm by a microplate reader (HT; BioTek, Winooski, VT, USA) (*Thawini & Al-Shawi, 2021*).

## Statistical analysis
SPSS analytics software (version 22), and Microsoft Office Excel 2020 for utilizing the statistical analysis of raw data. All values were evaluated using Graph Pad Prism (version 8.1). When applicable, suitable statistical procedures, such as one-way ANOVA followed by Tukey's post hoc test, were used to assess differences between variables of interest. The results were presented as mean ± standard deviation, and a *p*-value of less than 0.05 was deemed statistically significant. Three times the results were conducted.

**Table 1 Proximate chemical analysis of sidr plant.**

| Proximate chemical analysis (% D.W.) | Leaves | Pulps | Seeds |
|---|---|---|---|
| Moisture | $7.12 \pm 0.34^c$ | $12.25 \pm 0.47^a$ | $9.01 \pm 0.84^b$ |
| Ash | $8.17 \pm 0.33^a$ | $2.88 \pm 0.43^b$ | $0.4 \pm 0.06^c$ |
| Ether | $3.38 \pm 0.41^b$ | $0.6 \pm 0.08^c$ | $14.1 \pm 0.65^a$ |
| Protein | $12.44 \pm 0.46^a$ | $7.04 \pm 0.37^b$ | $6.99 \pm 0.42^c$ |
| Fat | $1.49 \pm 0.26^b$ | $0.46 \pm 0.09^c$ | $1.91 \pm 0.33^a$ |
| Carbohydrate | $70.88 \pm 1.59^b$ | $64.26 \pm 0.61^c$ | $81.69 \pm 2.23^a$ |
| Fiber | $14.1 \pm 0.86^b$ | $4.54 \pm 0.42^c$ | $49.22 \pm 1.24^a$ |

**Notes.**
 * The mean ± standard deviation is displayed for each value. Significant changes between the various portions of the sidr plant ($n = 3$) are indicated by different letters (a, b, c).

# RESULTS AND DISCUSSION

## Proximate chemical analysis

Some nutrients were evaluated in the sidr plant (leaves, pulps, and seeds), Table 1. The dry weight of sidr samples was used to evaluate the proximate chemical analysis. The results presented that leaves had the highest contents of ash and protein (8.17%, and 12.44%, respectively); pulps had the highest value of moisture content (12.25%); while seeds had the highest contents of ether, fat, carbohydrate, and fiber contents (14.1%, 1.91%, 81.69%, and 49.22%, respectively). According to the results, carbohydrate values were the major components of the total solids of sidr plant, which was linked with *Ahmed & Sati (2018)* who reported the carbohydrate contents in seeds 11.50–35.25% DW and pulps 58.02–82.70% DW. Meanwhile, carbohydrates help the body's metabolism. Seeds may be a good source of fiber and energy required for body functions. *Memon et al. (2012)* reported the composition of the seed oil which included high amounts of tocopherol, sterols, and monounsaturated fats which play an essential role in lowering cholesterol levels and heart disease risks. *Hashem & El-Lahot (2021)* reported similar values for ether values in dehydrated pulps and seeds. Measurable quantities of nutrients included in sidr plant make it suitable for enhancement as dietary supplements (*Abubakar et al., 2017*). The fiber contents of sidr leaves, which have pharmacological qualities including anti-inflammatory, anti-ulcer, and anti-allergic effects are important for the health system. While the indigestible cellulose which contains a sufficient amount of nutrients, can be used as a substitute resource for ruminant feed (*Mohd Jailani et al., 2020*). Varieties, genetic factors, ripening stages, geographical locations, harvesting times, climatic conditions, and environmental factors are some of the important elements that may be responsible for the differences in the proximate chemical analysis (*Al-Naimi et al., 2023*).

## Mineral elements

The existence of numerous mineral elements is frequently linked to ash contents, they are essential for evaluating the food's quality and nutritional content. Table 2 presents the mineral profile of the sidr parts. The comparative study showed that sidr plant is a reliable supplier of both macro and micro elements.; leaves had the lowest concentrations of Na (59.89 mg/100 g), Zn (1.09 mg/100 g), Cu (0.41 mg/100 g) with the highest values of Ca

**Table 2  Mineral contents of sidr plant.**

| Minerals (mg/100 g) | Leaves | Pulps | Seeds |
|---|---|---|---|
| Ca | 419.21 ± 8.88[a] | 340.85 ± 11.73[b] | 155.47 ± 4.13[c] |
| P | 195.17 ± 4.75[b] | 138.55 ± 3.99[c] | 204.47 ± 9.13[a] |
| Na | 59.89 ± 4.13[c] | 133.47 ± 5.02[b] | 160.11 ± 5.67[a] |
| K | 106.02 ± 3.26[b] | 866.14 ± 13.25[a] | 103.65 ± 5.67[c] |
| Mg | 145.78 ± 5.13[a] | 77.2 ± 2.33[b] | 40.02 ± 3.74[c] |
| Zn | 1.09 ± 0.11[c] | 13.15 ± 1.06[a] | 1.44 ± 0.21[b] |
| Cu | 0.41 ± 0.04[c] | 1.14 ± 0.12[a] | 1.04 ± 0.07[b] |
| Mn | 1.77 ± 0.21[b] | 17.25 ± 2.02[a] | 0.32 ± 0.06[c] |
| Fe | 38.02 ± 2.81[a] | 10.48 ± 1.37[c] | 19.78 ± 2.82[b] |
| Cr | 0.12 ± 0.07[a] | 0.1 ± 0.07[c] | 0.11 ± 0.06[b] |
| B | 0.51 ± 0.04[c] | 1.39 ± 0.28[a] | 0.22 ± 0.04[b] |

**Notes.**
* The mean ± standard deviation is displayed for each value. Significant changes between the various portions of the sidr plant ($n = 3$) are indicated by different letters (a, b, c).

(419.21 mg/100 g), Mg (145.78 mg/100 g), Fe (38.02 mg/100 g), and Cr (0.12 mg/100 g). The data presented that the pulps were the best source of several minerals such as Zn (13.15 mg/100 g), Cu (1.14 mg/100 g), Mn (17.25 mg/100 g), and B (1.39 mg/100 g). Potassium is described as a blood pressure lowering agent which presented (866.14 mg/100 g). *El Maaiden et al. (2020)* reported the mineral amounts in pulps were in the correct range of the reference intake (RIs) of the daily requirements. Iron is well recognized for the body to transmit oxygen, while pulps have higher iron contents than apples (0.12 mg/100 g) (*Kasuher et al., 2024*). Seeds had the highest mineral contents of P (204.47 mg/100 g) and Na (160.11 mg/100 g) with the low contents of Ca (155.47 mg/100 g), K (103.65 mg/100 g), Mn (0.32 mg/100 g), and B (0.22 mg/100 g). The fruit pulp's high calcium, (340.85 mg/100 g) makes the fruits considered as natural sources of supplementations of those elements for kids and nursing mothers (*Osman & Asif Ahmed, 2009*). Incorporating sidr leaves into the diet can contribute to meeting daily mineral requirements, especially for calcium, potassium, and iron (*Sarjito et al., 2021*; *Fatema & Nedaa was, 2024*). Variations in mineral contents can be due to several factors, including varieties, soil qualities, irrigation regimes, geographical sources, harvest periods, maturations, storage conditions, and ripeness within different sections of the same fruit uses (*Hashem & El-Lahot, 2021*).

## Physicochemical parameters and sugar compositions

The pH value ranged from 3.75 in pulps to 5.48 in leaves according to Table 3's physicochemical properties. The pH value of the sidr plant makes it ideal for the skin. Sidr leaf extract was utilized in an emulsion formulation that demonstrated cosmeceutical qualities like anti-aging, skin whitening, and moisturizing on human skin (*Akhtar et al., 2016*). Leaves had the highest values for total chlorophyll (3.3 mg/g) and ascorbic acid (41.78 mg/100 g) with the lowest values for reducing sugar (2.77%), TSS (0.98 Brix°), and TA (0.08%). Ascorbic acid is essential for biosynthesis and antioxidant processes as it enhances iron and tin absorption and immunological functions (*Sami et al., 2014*).

**Table 3  Physicochemical parameters of sidr plant.**

| Physicochemical parameters | Leaves | Pulps | Seeds |
|---|---|---|---|
| Total Chlorophyll (mg/g) | 3.3 ± 0.39[a] | 1.77 ± 0.15[b] | 0 ± 0[c] |
| Reducing Sugar (%) | 2.77 ± 0.21[c] | 44.25 ± 1.14[a] | 29.14 ± 2.18[b] |
| Starch (%) | 4.88 ± 0.41[b] | 16.59 ± 0.95[a] | 3.98 ± 0.53[c] |
| Ph | 5.48 ± 0.43[a] | 3.75 ± 0.58[c] | 4.05 ± 0.52[b] |
| TSS (Brix°) | 0.98 ± 0.15[c] | 11.66 ± 0.69[a] | 4.05 ± 0.53[b] |
| TA (%) | 0.08 ± 0.02[c] | 1.12 ± 0.53[a] | 1.01 ± 0.25[b] |
| Ascorbic Acid (mg/100 g) | 41.78 ± 1.46[a] | 19.02 ± 0.96[b] | 6.53 ± 0.45[c] |

Notes.
* The mean ± standard deviation is displayed for each value. Significant changes between the various portions of the sidr plant ($n = 3$) are indicated by different letters (a, b, c).

**Table 4  Sugar compositions of sidr plant.**

| Sugar compositions (mg/g) | Leaves | Pulps | Seeds |
|---|---|---|---|
| Sucrose | 8.81 ± 0.42[c] | 41.22 ± 2.65[a] | 9.84 ± 0.92[b] |
| Fructose | 0.5 ± 0.18[c] | 23.17 ± 1.94[a] | 1.56 ± 0.16[b] |
| Glucose | 0.38 ± 0.11[c] | 26.08 ± 1.56[a] | 4.41 ± 0.51[b] |

Notes.
* The mean ± standard deviation is displayed for each value. Significant changes between the various portions of the sidr plant ($n = 3$) are indicated by different letters (a, b, c).

The relationship between the Brix° value and the amount of sugar in the sample showed that leaves with low TSS levels have less sugar, making them less susceptible to bacterial development and treating infections (*Mohd Jailani et al., 2020*). However, seeds had slightly the lowest starch value (3.98%), and ascorbic acid content (6.53 mg/100 g), with no values for total chlorophyll. The reducing sugar (44.25%), starch (16.59%), TSS (11.66 Brix°), and TA (1.12%) values were slightly the highest in pulps.

Consequently, the fruit's high total carbohydrate content is the main reason for its sweet taste. Lactose, dextrose, maltose, and fructose were the main reducing sugars found in pulps and seeds, according to the previous study by *Amoo & Atasie (2012)*.

The results of sugar compositions are shown in Table 4. Pulps had the major values of (41.22, 23.17, and 26.08 mg/g), followed by seeds (9.84, 1.56, 4.41 mg/g), while leaves detected the lowest values (8.81, 0.50, 0.38 mg/g) for sucrose, fructose, and glucose, respectively. Results were linked with *Wang et al. (2022)* who studied comparative sugars among leaves and fruits. According to the previous results, sidr leaves can be considered natural corrosion inhibitors and sources for therapeutic products.

## Phenolics, flavonoids contents and antioxidant activities

Two different extraction yields of phenolics, flavonoid contents, and antioxidant activities are shwn in Tables 5 and 6. The results reported that the effectiveness of the aqueous extract afforded the highest phenolics and antioxidant activity (%) yields in all different parts of the sidr plant compared to ethanolic extract which had the highest effectiveness for flavonoids and antioxidant activity ($IC_{50}$) yields. Total phenols ranged in aqueous extract from 18.69 mg GAE/g in leaves to 45.01 mg GAE/g in pulps. Sidr plant has a significant concentration

**Table 5  Phenolics, flavonoids contents and antioxidant activities of the sidr ethanolic extract.**

| Ethanolic extract | Leaves | Pulps | Seeds |
|---|---|---|---|
| Total Phenol (mg GAE/g) | 12.74 ± 0.85[c] | 21.54 ± 2.33[a] | 17.32 ± 0.82[b] |
| Total Flavonoid (mg QE/100 g) | 111.65 ± 1.97[c] | 178.45 ± 4.34[a] | 168.14 ± 2.42[b] |
| Antioxidant Activity DPPH (%) | 80.65 ± 2.84[a] | 40.55 ± 3.09[b] | 35.89 ± 1.74[c] |
| DPPH ($IC_{50}$) (mg/mL) | 0.5 ± 0.17[c] | 12.01 ± 0.58[b] | 13.52 ± 1.07[a] |

Notes.
*The mean ± standard deviation is displayed for each value. Significant changes between the various portions of the sidr plant ($n = 3$) are indicated by different letters (a, b, c).

**Table 6  Phenolics, flavonoids contents and antioxidant activities of the sidr aqueous extract.**

| Aqueous extract | Leaves | Pulps | Seeds |
|---|---|---|---|
| Total Phenol (mg GAE/g) | 18.69 ± 1.46[c] | 45.01 ± 2.23[a] | 44.8 ± 2.95[b] |
| Total Flavonoid (mg QE/100 g) | 102.14 ± 2.55[c] | 163.58 ± 3.46[a] | 103.74 ± 3.23[b] |
| Antioxidant Activity DPPH (%) | 83.09 ± 2.84[a] | 66.75 ± 2.51[b] | 38.11 ± 1.19[c] |
| DPPH ($IC_{50}$) (mg/mL) | 0.11 ± 0.06[c] | 7.66 ± 0.88[b] | 13.02 ± 0.67[a] |

Notes.
* The mean ± standard deviation is displayed for each value. Significant changes between the various portions of the sidr plant ($n = 3$) are indicated by different letters (a, b, c).

of phenolic chemicals, which can scavenge free radicals, exhibit redox characteristics, and provide antioxidant activity (*Elhady et al., 2024*). On the same trend, the ethanolic extract of pulps contained the highest flavonoid contents (178.45 mg QE/100 g) compared to leaves (111.65 mg QE/100 g). Phenolic and flavonoid components can be crucial in absorbing and scavenging free radicals, thereby slowing the development of cancers, cardiovascular, inflammatory, and neurodegenerative (*Tungmunnithum et al., 2018*). Those results were linked with *El Maaiden et al. (2019)* who used various extraction yields for the sidr plant. The highest antioxidant activities were detected in leaves which reported 80.65% and 83.09% for aqueous and ethanolic extracts with $IC_{50}$ values of 0.5 mg/mL and 0.11 mg/mL, respectively. The aqueous extract had higher values than the ethanolic extract in pulps as it presented 66.75% with an $IC_{50}$ value of 7.66 mg/mL compared with seeds which presented 38.11% with an $IC_{50}$ value of 13.02 mg/mL. Therefore, all parts of the sidr plant can be used as functional foods that enhance health due to their medical, cosmetic, pharmacological, and nutraceutical properties.

## Inhibitory effects on the bacterial growth

The results for the antibacterial activities of aqueous and ethanolic extracts against six popular food-borne bacterial strains were examined by the agar-well diffusion test, Figs. 2 and 3. The ethanolic extract showed inhibition zones, exhibiting more than 20 mm maximal inhibition zone against *Staphylococcus aureus* (21.47 mm) and *Bacillus subtilis* (24.66 mm) for leaves. The ethanolic extract for pulps presented higher inhibitory effects than seeds ranging (17.52 mm and 19.26 mm); (13.28 mm and 13.58 mm) against *Staphylococcus aureus* and *Bacillus subtilis*, respectively.

However, the least inhibitory effects were noted less than eight mm inhibition zone with the aqueous extract against *Escherichia coli* (7.33 mm), *Pseudomonas aeruginosa* (4.02 mm),

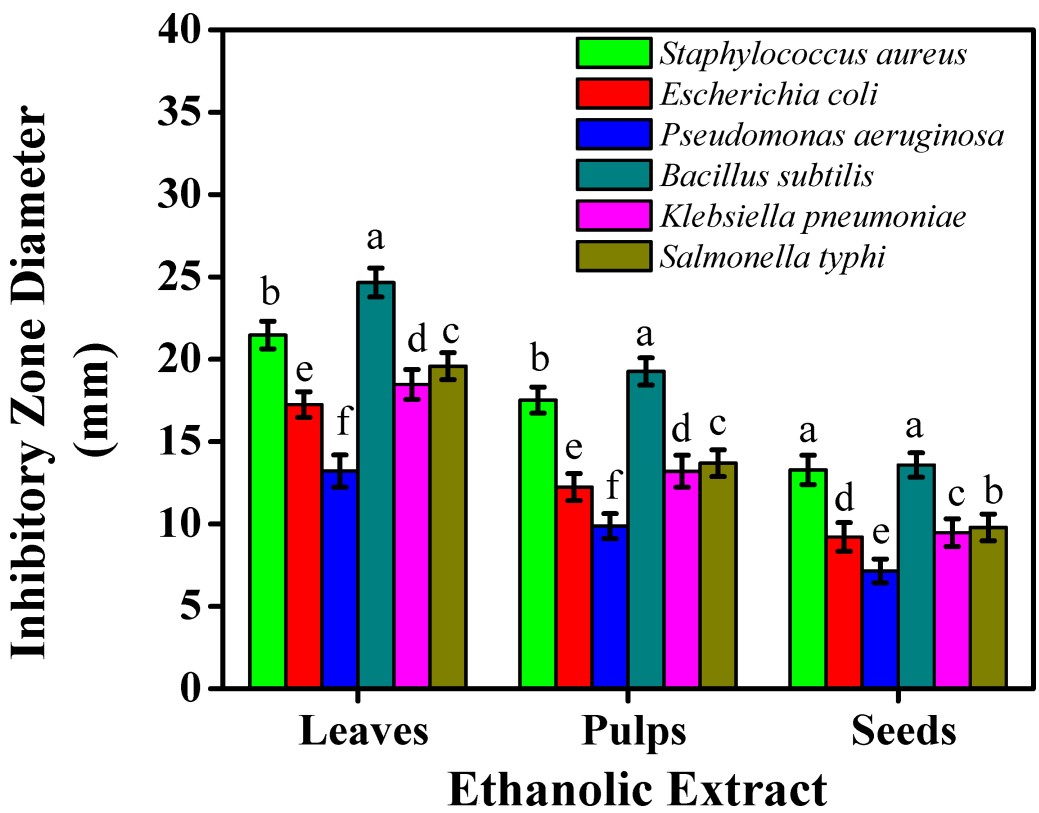

**Figure 2** The inhibitory effects on the bacterial growth of the sidr ethanolic extract.

*Klebsiella pneumoniae* (7.45 mm), and *Salmonella typhi* (7.19 mm) for seeds. The ethanolic extract against *Pseudomonas aeruginosa* reported the least inhibitory effect (7.15 mm) for seeds. Aqueous extract for leaves presented higher inhibitory effects than pulps and seeds which were reported (15.14 mm, 19.36, and 14.22 mm) against *Staphylococcus aureus*, *Bacillus subtilis*, and *Salmonella typhi*, respectively.

Results on the inhibitory effects on the bacterial growth on leaves were linked with *Salih et al. (2019)*, who studied three species of sidr leaves against five bacterial strains. *Taha et al. (2021)*, studied the presence of squalene, a steroid class member found in leaf extracts which can act as an antioxidant and antibacterial agent against *Pseudomonas aeruginosa*. These results showed that sidr's ethanolic extract was typically more effective than its aqueous extract at preventing bacterial growth. This might be due to the presence of some active components which improved the solubility in ethanol and increased their accessibility and capacity to interact with bacterial strains. According to the microbial assessments, sidr leaves can be used in treating a variety of infections due to the presence of glycoside, terpenoid, saponin, steroid, tannin, sapogenin, flavonoid, cyclopeptide alkaloid, leucocyanidin, resin, furanocoumarin, triterpenoids, and phenolic components, whose antimicrobial and antioxidant properties are well known (*Ali, Almagboul & Mohammed, 2015*). Instead of a single bioactive component, a mixture of these secondary metabolites is usually responsible for the positive therapeutic effects of plant materials (*Jain, Khatana & Vijayvergia, 2019*).

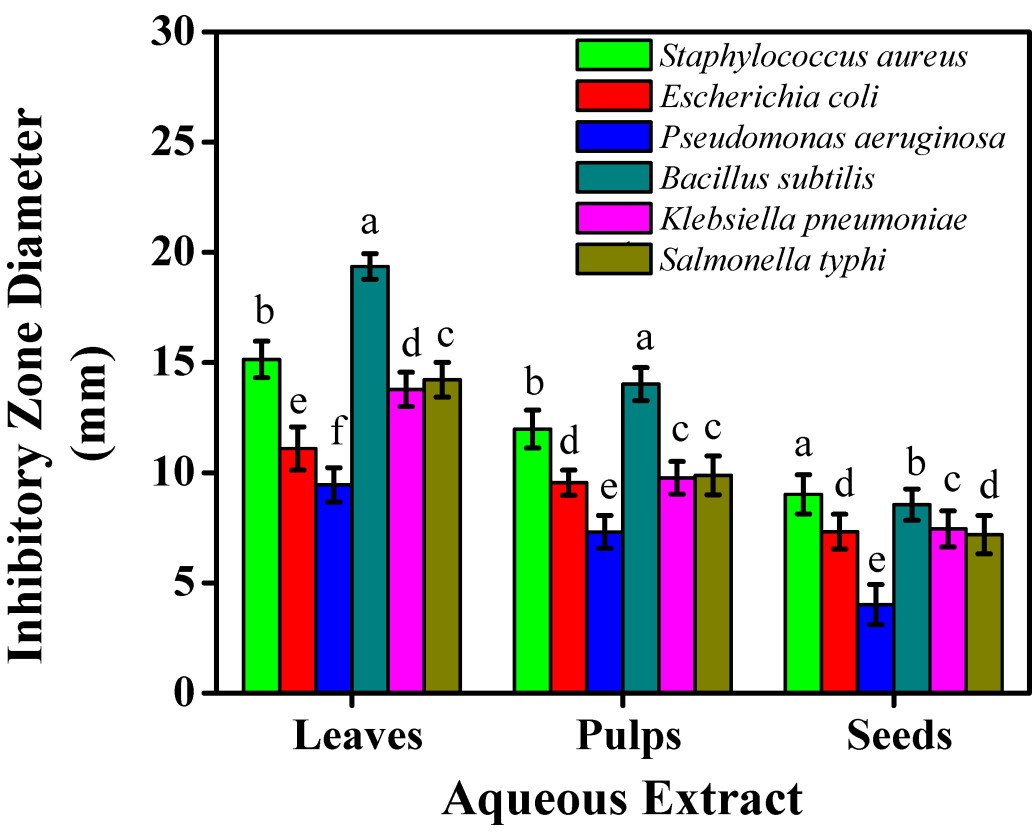

**Figure 3** The inhibitory effects on the bacterial growth of the sidr aqueous extract.

The process of preventing microbial development entails breaking down the integrity breaching the cytoplasmic cell membrane of bacteria and resulting in intracellular leakage. Cell death is the result of the accumulation of hydrophobic groups on phospholipids, while cell DNA and RNA synthesis can be inhibited by phenolic components. Furthermore, the terpenoid group can hinder microbial growth and impair the effectiveness of cell membranes. Cis-9 hexadecenal was the most important antibacterial component found in the side leaf extract (*Sarjito et al., 2021*). The sidr plant has a variety of medicinal uses, including treating diabetes, reducing high blood pressure, disinfecting wounds, and inhibiting several bacterial and fungal pathogens (*Albalawi, 2021*). Since resistant strains of bacteria have been created as a result of antibiotic overuse, the quest for compounds with strong antibacterial qualities is an important area of research. Since plant-derived chemicals are inherently hazardous to bacteria but not to humans, they present a possible substitute to create natural antimicrobial drugs, which may aid in tackling the escalating problem of antibiotic resistance (*AlSheikh et al., 2020*). Sidr plants could serve as a natural antibacterial supply, especially against bacterial species. Standardized formulations for therapeutic use can be developed using the effective extracts found in this investigation as a basis.

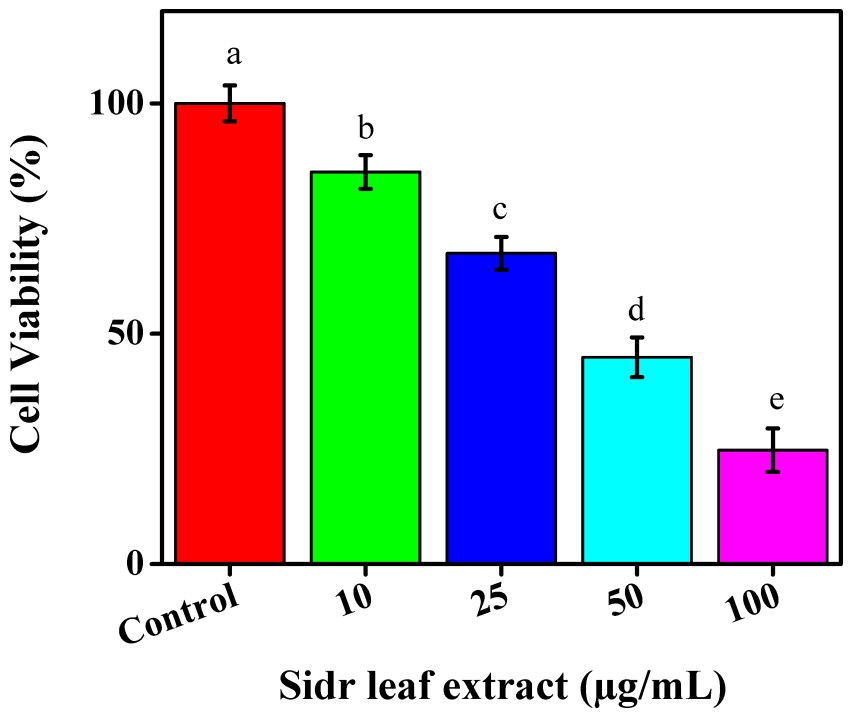

**Figure 4** The cytotoxic effects of sidr leaves on breast cancer cells (MCF-7).

## Cytotoxic effects on breast cancer cells

One of the leading causes of death is cancer. Cancer can be prevented by diet and lifestyle choices. Consequently, there is a lot of interest in finding chemotherapeutic or even chemopreventative drugs in safe and natural goods (*Zahara et al., 2022*). Many studies were presently being conducted to develop ecologically friendly production methods for cancer treatment (*Al-Eisa et al., 2023*). This study evaluated the ethanolic extract for leaves' potential for toxicity in the MCF-7 cell line using the standard MTT assay. A 96-well plate was used for different doses of ethanolic extract for leaves as 10, 25, 50, and 100 µg/mL for 24 h. Moreover, the various extracts reduced the MCF-7 cell line's viability in a concentration-dependent manner to reach 85.11%, 67.45%, 44.89%, and 24.72%, respectively, Fig. 4.

The results were contrasted with control cells, which exhibit dosage-dependent inhibition. The findings of the MTT assay showed that the ethanolic extract of leaves inhibited MCF-7 cell viability by >50% at 100 µg/mL after 24 h. The results showed that cell viability decreased with the ethanolic extract for leaves concentration dose, indicating that additional concentration can accumulate inside cells, causing stress and ultimately cell death. Guidelines for cancer therapy are poor, and the disease now has a high fatality rate worldwide (*Thawini & Al-Shawi, 2021*).

## Reactive oxygen species generation

As signaling molecules, ROS are essential and have been shown to have anti-tumorigenic actions. They preserve cellular homeostasis and perform essential signaling tasks at low

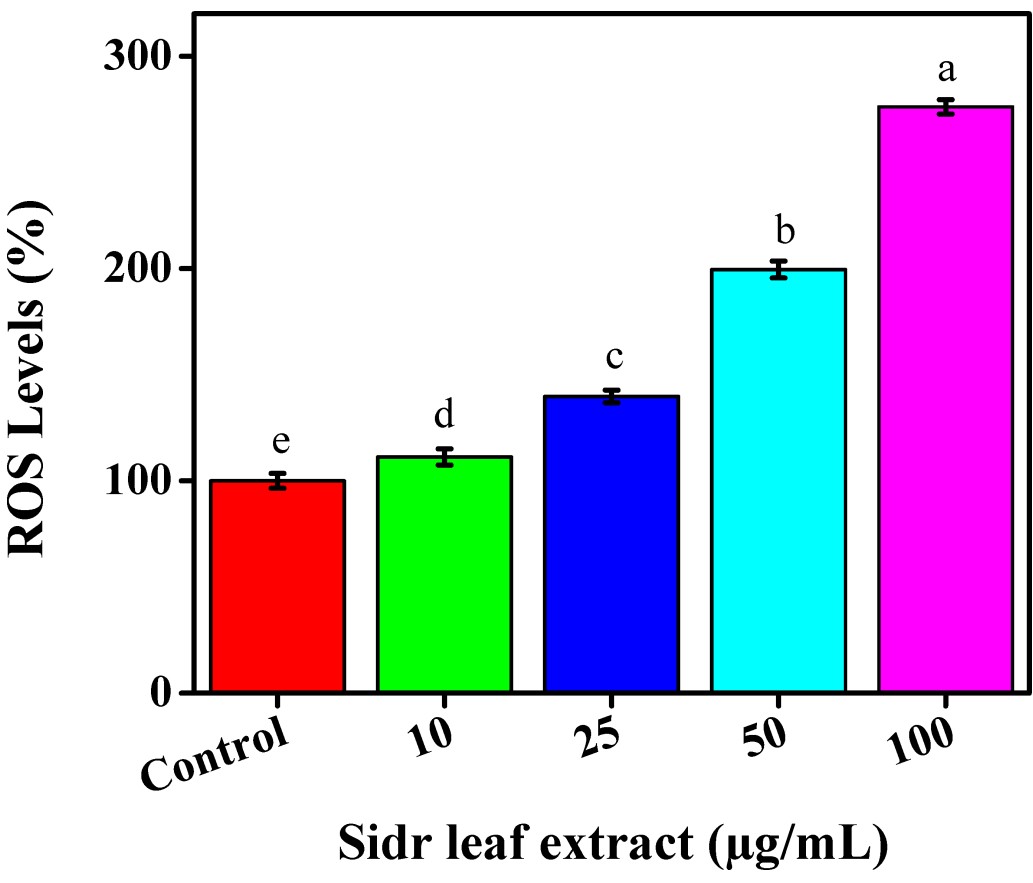

**Figure 5** **Reactive oxygen species generation of sidr leaves on breast cancer cells (MCF-7).**

concentrations. When ROS present in high concentrations, they can harm lipids, proteins, and genomic DNA, among other biological molecules, which can result in the development of tumors. ROS have also been linked to tumor invasion and metastasis. Conversely, elevated ROS levels can prevent tumor growth by killing cancer cells and stopping their proliferation. The primary cause of ROS excess is recognized to be mitochondrial malfunction (*Alshahrani et al., 2024*). Reactive oxygen species were produced by the MCF-7 cell line following a 24-hour exposure to the ethanolic extract for leaves and raised the amounts of ROS inside the cells in a concentration-dependent manner, as seen by the rise in DCFDA fluorescence. When ethanolic extracts were exposed to a DCFH-DA kit (0, 10, 25, 50, and 100 μg/mL), the MCF-7 cell line showed a slightly increased amount to reach 100%, 111.25%, 139.78%, 199.54%, and 276.15% respectively. These findings imply that the ethanolic extract for leaves at least partially inhibits TNBC cell proliferation *via* a ROS-dependent mechanism, Fig. 5.

Anticancer medications that directly accumulate ROS are effective in treating several cancer types, but because they impact both cancer and healthy cells, their effects on healthy cells are still debatable (*Thawini & Al-Shawi, 2021*). Results were in agreement with *Abdallah et al. (2024)* who examined other techniques as intrinsic apoptotic pathways,

and autophagy activation against TNBC viability. Another prior research by *Mesmar et al. (2021)* demonstrated that the anti-proliferative impact of sidr plant for pancreatic cancer cells in humans was affected by ROS inhibition.

Although the experiments revealed significant antibacterial, antioxidant, and anticancer properties in the ethanolic leaf extract of *Ziziphus spina-christi*, we recognize that these results showed correlation rather than direct causation. The antioxidant and antibacterial qualities of the extract may contribute, but not necessarily cause, the anticancer effects observed in MCF-7 cells. Causal relationships between these bioactivities cannot be definitively demonstrated in the absence of direct mechanistic data, such as pathway analysis, gene expression investigations, or specific inhibitor trials.

## CONCLUSION

The primary goal of the current investigation was to determine the nutrients, phytochemical composition, and antimicrobial assessments for various sidr parts and highlight the anticancer effects on MCF-7 breast cells. Therefore, all parts of the sidr plant can be used as functional foods that enhance health due to their medical, cosmetic, pharmacological, and nutraceutical properties. Sidr plants could serve as a natural antibacterial supply, especially against bacterial species. Standardized formulations for therapeutic use can be developed using the effective extracts found in this investigation as a basis. Generally, the current work indicates that sidr plant can be used as a medicinal agent to treat breast cancer cells. It is advised that future research investigate clinical uses and enhance extraction procedures for pharmaceutical development that is scalable.

### Funding
This research was funded by Taif University, Saudi Arabia, Project No. (TU-DSPP-2024-10). The funders had no role in study design, data collection and analysis, decision to publish, or preparation of the manuscript.

### Grant Disclosures
The following grant information was disclosed by the authors:
Taif University, Saudi Arabia: TU-DSPP-2024-10.

### Competing Interests
The authors declare there are no competing interests.

### Author Contributions
- Roqayah H. Kadi conceived and designed the experiments, prepared figures and/or tables, and approved the final draft.
- Nashi K. Alqahtani conceived and designed the experiments, prepared figures and/or tables, and approved the final draft.
- Ahmed M. Abdulfattah performed the experiments, prepared figures and/or tables, and approved the final draft.

- Fayez Alsulaimani analyzed the data, authored or reviewed drafts of the article, and approved the final draft.
- Ahmed M. Basri conceived and designed the experiments, prepared figures and/or tables, and approved the final draft.
- Reham M. Algheshairy conceived and designed the experiments, prepared figures and/or tables, and approved the final draft.
- Hend F. Alharbi performed the experiments, authored or reviewed drafts of the article, and approved the final draft.
- Rokayya Sami conceived and designed the experiments, prepared figures and/or tables, and approved the final draft.
- Amal Alyamani conceived and designed the experiments, prepared figures and/or tables, and approved the final draft.
- Roua S. Baty analyzed the data, authored or reviewed drafts of the article, and approved the final draft.
- Ruqaiah I. Bedaiwi analyzed the data, authored or reviewed drafts of the article, and approved the final draft.
- Hala M. Abo-Dief performed the experiments, prepared figures and/or tables, and approved the final draft.
- Nahid A. Osman analyzed the data, authored or reviewed drafts of the article, and approved the final draft.
- Nouf H. Alsubhi analyzed the data, authored or reviewed drafts of the article, and approved the final draft.
- Ashwaq M. Al-Nazawi performed the experiments, prepared figures and/or tables, and approved the final draft.
- Manal Malibary performed the experiments, prepared figures and/or tables, and approved the final draft.

## Data Availability

The raw data is available in the Supplemental Files.

## Supplemental Information

Supplemental information for this article can be found online at http://dx.doi.org/10.7717/peerj.19858#supplemental-information.

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
