# Peer review of "Anticancer potential of ethanolic sidr leaf extracts against MCF-7 breast cancer cells: phytochemical, nutritional, and antimicrobial profile comparisons of different plant parts"

_PeerJ, doi:10.7717/peerj.19858_

## Round 0.1 · original submission · Major Revisions

Please, authors, kindly attend to the reviewers comments.

**Language Note:** The review process has identified that the English language must be improved. PeerJ can provide language editing services - please contact us at [email protected] for pricing (be sure to provide your manuscript number and title). Alternatively, you should make your own arrangements to improve the language quality and provide details in your response letter. – PeerJ Staff

·

Basic reporting

The article entitled “Anticancer effects of sidr plant on MCF-7 breast cells with a comparative study for nutrients, phytochemical composition, and antimicrobial assessments” by Roqayah H. Kadi et al. is a very interesting work centering on the anticancer effect of different components of sidr plant as they are also being analysed.
The subject is appealing scientifically having not only a health virtue in the fight against one of the most alarming diseases in the world (cancer) but also an industrial interest seeing that the sidr plant Ziziphus spina-christi (L.) is among other countries (India, Pakistan, Ethiopia, Egypt, Libya, Sudan) native of Saudi Arabia (Naghmouchi & Alsubeie, 2021).
Beginning with the title it may clarify the fact that the possible neutraceutical-health benefits – anticancer and antibacterial properties are studied according to the macro-micro nutrient- phytochemical content and physico-chemical virtues. The study is not only centered on the anticancer effect as the readers discover.
The abstract contains certain syntax mistakes, with more specifically:
Line 49: sidr was a good source of… not sidr had….
Line 59: with the phrase starting with generally which is not correct in meaning; generally must be replaced with :”in conclusion”. To note also that any abbreviation used must be previously explained un the text.
All through the scientific document there are syntax mistakes that must be corrected to ensure the fluidity of the expression and facilitate comprehension of the reader.
The abstract must be clearer vis a vis of the methodology of the study.
Having an abstract with the following subheadings may be clearer for readers : Backround, Methodology, results and conclusion.
The authors are urged to correct the wording line 72 when describing cancer; seeing that the term “mass of cell cycle” … “up normal cell growth” is incorrect scientifically; cancer cell growth is uncontrollable.
The references are interesting and more importantly recent.
Regarding the investigation as a whole, it is essential to explain why the ethanolic leaf extracts specifically are the potent anticancer part of the plant used in the project from other work highlighting their possible anticancer effect.
In the methodology, statistical analysis must mention not only the types of software used but more importantly the types of tests conducted between the variables of interest.
Regarding the different chemical tests performed throughout the study, it is necessary to refer to figure 1 more clearly that summarizes the serious work conducted in phases.
In the results, the authors comment on the macro and micronutrient typology and content of different parts of the sidr plant analyzed; in order to refer to possible nutraceutical effects, it is key to discuss the quantities of the specific nutrients mentioned and proven possible health activities reported in the scientific literature. In this specific article the focus should be more on the anticancer effect of the sidr plant and it’s nutrients/phytochemical makeup also.
Beneath each table, the statistical test used to compare data must be mentioned.
Regarding the histograms the numbers are mean with standard deviations or standard error mean?
Line 278, the ph value can be applied topically should be changed to the ph value of the sidr plant makes it use ideal for skin etc… As mentioned previously the English syntax must be revised, please. Important to note that mistakes are normal when we write.
Line 322: six kings of food-borne bacteria … the term king is not appropriate; replace with six main or popular food-borne bacteria…
Line 381 the whole phrasing must be changed du to ambiguity of the wording, if possible, explain what plant derived proteins are interesting … and what cofactors?
I advise authors to conclude their work with future studies to be conducted and not a statement that in itself deserves further studies to be proven.

Experimental design

The result tables require the statistical tests used to be mentionned
The histograms must mention the type of data displayed, if mean / average with standard deviation or SEM.

The statistical analysis must mention the types of tests used on the variables studied.

Validity of the findings

Regarding the investigation as a whole, it is essential to explain why the ethanolic leaf extracts specifically are the potent anticancer part of the plant used in the project from other work highlighting their possible anticancer effect.

The work seems to be more health effect than the anticancer effect that is highlighted more than it should be ; reason why the title should change.

Reviewer 2 ·

Basic reporting

Dear authors,
This work is preliminary. There is not any mechanistic insight. Therefore, I suggest you add mechanisms of action, either apoptotic cell death, cell cycle arrest, or survival signaling inhibition.

Experimental design

Need to add mechanism of assay..

Validity of the findings

Not up to mark

Additional comments

Not applicable

·

Basic reporting

• Lang. editing : Awkward Phrasing: Constructions such as “dose of more than 50 % of MCF-7 cells” are unclear—better phrased as “inhibited MCF-7 cell viability by > 50 % at X µg/mL.”]
Inconsistent Tense & Voice: Shifts between past and present tense (“The results presented that…” vs. “Results show that…”) and passive/active voice lead to confusion.
I recommend a thorough copy‐edit by a native-English–proficient editor or professional service to correct these issues and ensure consistency of style, grammar, and terminology
.
• Introduction & Context: Provide a clear statement of the study’s novelty, why comparing leaves, pulp, and seeds yields new insights and explicitly articulate the knowledge gap.
• Citations & References: Standardize reference formatting (remove duplicated/misformatted DOIs) and ensure all factual claims (e.g., breast cancer incidence) cite current, authoritative sources.
• Figures & Tables: Embed high-resolution figures with properly labeled axes, units, and error bars; annotate statistical significance. In tables, use superscript letters (a, b, c) with a legend explaining statistical tests.
• Data Availability: Include a supplementary index linking raw data files to each experiment to enhance reproducibility.

- Literature Citations: Please enrich your Introduction and Discussion by citing recent work on Ziziphus spina-christi. For example, Mahmoud A.H. Mostafa, Hani M.J. Khojah & Tomihisa Ohta (2023) report the isolation of antitumor constituents sidrin and sidroside from Z. spina-christi (Saudi Pharm. J. 31(6): 1019–1028; https://doi.org/10.1016/j.jsps.2023.04.029). Including this will strengthen your rationale and contextualize your anticancer findings.

Experimental design

• Hypothesis & Objectives: State a testable hypothesis regarding expected differences among plant parts in the Introduction.
• Authentication: Supply voucher specimen details (collection site, herbarium accession number) for plant material.
• Extraction Details: Report extraction yields (%) for each solvent and plant part, and specify sample weights and solvent volumes.
• Replicates & Statistics: Clarify that assays were performed in biological triplicate (n = 3), present data as mean ± SD, and specify statistical tests (e.g., one-way ANOVA with Tukey’s post hoc).
• Controls: Justify concentration choices in antimicrobial tests or include dose–response data; for cell assays, detail vehicle controls and include a positive control (e.g., H₂O₂) in ROS measurements.

- Antimicrobial Methods: In your Methods section, please reference established protocols for plant-extract antimicrobial testing. For instance, Khojah H.M.J., Abdelhalim O.B., Mostafa M.A.H. & Habib E.E. (2022) describe such assays on desert truffle extracts (Saudi J. Biol. Sci. 29(11): 103462; https://doi.org/10.1016/j.sjbs.2022.103462). Adapting their positive- and negative-control setup will improve comparability and methodological rigor.

Validity of the findings

• Data Presentation: Present full dose–response curves for viability and ROS assays with error bars and IC₅₀ values (with 95 % confidence intervals).
• Mechanistic Claims: Either temper assertions of ROS-mediated cytotoxicity or add mechanistic assays (e.g., apoptosis markers, caspase activation) to substantiate the claim.
• Interpretation: Avoid implying causation between antimicrobial/antioxidant activity and anticancer effects without direct evidence; discuss results in the context of similar Ziziphus studies, noting any discrepancies.

Additional comments

General Comments
• Strengths: The multi-angle comparison (nutrients, phytochemicals, antimicrobial, anticancer) and use of two extraction solvents provide a rich dataset.
• Major Recommendations:
1. Secure professional language editing.
2. Add methodological details (authentication, extraction yield, replicates).
3. Improve statistical reporting and figure/table annotation.
4. Moderate overreaching conclusions or include additional mechanistic data.
• Minor Suggestions: Standardize unit formatting (°C, µg/mL), define all acronyms at first use (e.g., TSS, TA), and correct spelling errors (e.g., “Assessments”).

---

## Round 0.2 · Minor Revisions

Please kindly address the comments raised. Thank you

**Language Note:** The review process has identified that the English language must be improved. PeerJ can provide language editing services - please contact us at [email protected] for pricing (be sure to provide your manuscript number and title). Alternatively, you should make your own arrangements to improve the language quality and provide details in your response letter. – PeerJ Staff

·

Basic reporting

The english still requires some fine tuning to be honest before final submission.
To mention in the manuscript the term "serious work" is not advisable; thus please delete.

Experimental design

the overall English requires rereading one final time please.

Validity of the findings

the findings remain interesting and a stepping stone for further analysis

Additional comments

no additional comments

---

## Round 0.3 · Minor Revisions

Thank you authors for your patience.

Although reviewer considers your revised manuscript acceptable for publication, the editor encourages you to please address the following points to further elevate the quality of this scholarly piece of work.

a) Please kindly revise the title to: "Anticancer Potential of Ethanolic Sidr Leaf Extracts Against MCF-7 Breast Cancer Cells: Phytochemical, Nutritional, and Antimicrobial Profile Comparisons of Different Plant Parts"
This title better captures the work. Please, make sure to reflect this in the objective statement, starting from the abstract, to the last paragraph of the introduction.
The current investigation aimed to understand the potential therapeutic properties of the sidr plant against TNBC on MCF-7 breast cell line by conducting phytochemical, nutritional, and antimicrobial profile comparisons of different plant parts.

b) Please revise the abstract to strictly follow this:

Background and objective
Traditional medicine has long utilized natural plants to treat diseases and promote overall health. They have contributed significantly to the creation of modern medications. To supplement existing information, therefore, this current investigation aimed to understand the potential therapeutic properties of the sidr plant against TNBC on MCF-7 breast cell line by conducting phytochemical, nutritional, and antimicrobial profile comparisons of different plant parts.
Methods
Phytochemical analyses involved...., whereas nutritional analysis involved..... Additionally, antimicrobial analyses involved .....
Results
(Please shorten this to three short sentences, condense it further)
Conclusion
Bioactive components, antioxidant, antimicrobial and nutritional elements of sidr plant appears a promising medicinal candidate to fight breast cancer cells.
(Please make sure that this specific conclusion sentence is repeated somewhere in the conclusion section of this work)

c) Introduction: Please, authors, it is important you include more information to strengthen the introduction. Create a new paragraph that discuss the following:
New Paragraph 2 (between lines 103-104): Tell us about the major plant extraction methods, make sure to mention traditional methods like maceration, percolation, decoction, and Soxhlet extraction, as well as modern methods like ultrasound-assisted extraction, microwave-assisted extraction, supercritical fluid extraction, and pressurized liquid extraction. Give more emphasis to the traditional, and why it is relevant today. Narrow it down to ethanolic methods, and why it is of increasingly standard practice in institutional chemistry labs Saudi Arabia.

Your Paragraph 3 should merge lines 104-119 as one paragraph. Please, there are some sentences in the paragraph 3 that need references. Make sure to capture the references.

d) Materials and methods should start a new subsection called:
"Schematic overview of the experimental program"
This must comprise at least 4 sentences. Sentence one should introduce the Figure 1 as schematic overview of the experimental program, showing .....(identify the basic steps). Sentence 2 should connect this diagram with the objective of this work. Sentence 3 should mention that all experimental procedures adhered to good laboratory practices prescribed at the (name of the lab, department, faculty and university, state, country).

Please, make sure you delete line 142 sentence.

e) The remaining aspect of materials and methods are fine

f) I am very happy with the results and discussion. It is very good

g) Conclusion needs more reflection from the authors. Please, tell us more about lessons to take home, and its practicality (All of you should brainstorm and identify these aspects). Also, where and what is the direction for future research, provide us some information

Please, make sure to do a very diligent revision, and incorporate all above-requested points.

Look forward to your revised scholarly manuscript.

Blessed regards

·

Basic reporting

ok

Experimental design

ok

Validity of the findings

ok

Additional comments

The scientific name of the plant is missing. Please see line 122 in the PDF (or line 129 in the Word file). Alternatively, you may wish to incorporate it earlier, such as on line 90 of the introduction, by referring to it as sider (Ziziphus spina-christi).in italic, Also consider adding this scientific name to the keywords to enhance discoverability.

---

## Round 0.4 · Minor Revisions

The revising was rushed and many aspects were ignored.

Please revise the abstract to strictly follow this:

Background and objective
Traditional medicine has long utilized natural plants to treat diseases and promote overall health. They have contributed significantly to the creation of modern medications. To supplement existing information, therefore, this current investigation aimed to understand the potential therapeutic properties of the sidr plant against TNBC on MCF-7 breast cell line by conducting phytochemical, nutritional, and antimicrobial profile comparisons of different plant parts.

Methods (you repeated the objective in the abstract. )
Phytochemical analyses involved...., whereas nutritional analysis involved..... Additionally, antimicrobial analyses involved ..... (provide details here)


c) Introduction: Provide more information to strengthen the introduction. Create a new paragraph that discuss the following:
New Paragraph 2 (between lines 103-104): Tell us about the major plant extraction methods, make sure to mention traditional methods like maceration, percolation, decoction, and Soxhlet extraction, as well as modern methods like ultrasound-assisted extraction, microwave-assisted extraction, supercritical fluid extraction, and pressurized liquid extraction. Give more emphasis to the traditional, and why it is relevant today. Narrow it down to ethanolic methods, and why it is of increasingly standard practice in institutional chemistry labs Saudi Arabia.
You did not provide adequate information

Your Paragraph 3 should merge lines 104-119 as one paragraph. Please, there are some sentences in the paragraph 3 that need references. Make sure to capture the references.

d) Materials and methods should start a new subsection called:

(You ignored this section...provide the needed information)***

"Schematic overview of the experimental program"
This must comprise at least 4 sentences. Sentence one should introduce the Figure 1 as schematic overview of the experimental program, showing .....(identify the basic steps). Sentence 2 should connect this diagram with the objective of this work. Sentence 3 should mention that all experimental procedures adhered to good laboratory practices prescribed at the (name of the lab, department, faculty and university, state, country).

e) The remaining aspect of materials and methods are fine

f) I am very happy with the results and discussion. It is very good

Please, make sure to do a very diligent revision, and incorporate all above-requested points.

---

## Round 0.5 · accepted · Accept

I confirm authors have addressed all concerns, and this current revised version is acceptable for publication. You all have done a very great work.

Thank you authors for finding PeerJ Life & Environment as your journal of choice, and look forward to your future scholarly contributions.

Congratulations and very best regards.